# Leprosy in a prison population: A new active search strategy and a prospective clinical analysis

**Fred Bernardes Filho**[1◉], **Jaci Maria Santana**[1◉], **Regina Coeli Palma de Almeida**[1◉], **Glauber Voltan**[1◉], **Natália Aparecida de Paula**[1◉], **Marcel Nani Leite**[1◉], **Claudia Maria Lincoln Silva**[1◉], **Camila Tormena**[1◉], **Lean Basoli**[1◉], **Joelma Menezes**[2◉], **Moises Batista da Silva**[3◉], **John Stewart Spencer**[4◉], **Wilson Marques, Jr**[5◉], **Norma Tiraboschi Foss**[1◉], **Marco Andrey Cipriani Frade**[1◉] *

**1** Dermatology Division, Department of Medical Clinics, Ribeirão Preto Medical School, University of São Paulo, Ribeirão Preto, São Paulo, Brazil, **2** Center of Penitentiary Progression of Jardinópolis, Penitentiary Administration Secretariat, Jardinópolis, São Paulo, Brazil, **3** Spatial Epidemiology Laboratory, Federal University of Pará, Castanhal, Pará, Brazil, **4** Colorado State University, Department of Microbiology, Immunology and Pathology, Fort Collins, Colorado, United States of America, **5** Department of Neuroscience and Behavioral Sciences, Ribeirão Preto Medical School, University of São Paulo, Ribeirão Preto, São Paulo, Brazil

◉ These authors contributed equally to this work.
* mandrey@fmrp.usp.br

**Data Availability Statement:** All relevant data are within the manuscript.

**Funding:** This work was supported by WHO Implementation Research Team of Ribeirão Preto

## Abstract

### Background

This study evaluates an active search strategy for leprosy diagnosis based on responses to a Leprosy Suspicion Questionnaire (LSQ), and analyzing the clinical, immunoepidemiological and follow-up aspects for individuals living in a prison population.

### Methods

A cross-sectional study based on a questionnaire posing 14 questions about leprosy symptoms and signs that was distributed to 1,400 prisoners. This was followed by dermatoneurological examination, anti-PGL-I serology and RLEP-PCR. Those without leprosy were placed in the Non-leprosy Group (NLG, n = 1,216) and those diagnosed with clinical symptoms of leprosy were placed in the Leprosy Group (LG, n = 34).

### Findings

In total, 896 LSQ were returned (64%), and 187 (20.9%) of the responses were deemed as positive for signs/symptoms, answering 2.7 questions on average. Clinically, 1,250 (89.3%) of the prisoners were evaluated resulting in the diagnosis of 34 new cases (LG), based on well-accepted clinical signs and symptoms, a new case detection rate of 2.7% within this population, while the NLG were comprised of 1,216 individuals. The confinement time medians were 39 months in the LG while it was 36 months in the NLG (p>0.05). The 31 leprosy cases who responded to the questionnaire (LSQ+) had an average of 1.5 responses. The symptoms "anesthetized skin area" and "pain in nerves" were most commonly mentioned in

Medical School; the Center of National Reference in Sanitary Dermatology focusing on Leprosy of Ribeirão Preto Clinical Hospital, Ribeirão Preto, São Paulo, Brazil; the Brazilian Health Ministry (MS/FAEPAFMRP-USP: 749145/ 2010 and 767202/ 2011); FIOCRUZ RIBEIRÃO PRETO - TED 163/ 2019 - Processo: N˚ 25380.102201/2019-62/ Projeto Fiotec: PRES-009-FIO-20; Fulbright Scholar to Brazil awards 2015-2016 and 2019-2020 (JSS); The Heiser Program of the New York Community Trust for Research in Leprosy (JSS, MBS) grants P15-000827, P16-000796 and P18-000250. The funders had no role in study design, data collection and analysis, decision to publish, or preparation of the manuscript.

**Competing interests:** The authors have declared that no competing interests exist.

the LG while "tingling, numbness in the hands/feet", "sensation of pricks and needles", "pain in nerves" and "spots on the skin" responses were found in more than 30% of questionnaires in the NLG. Clinically, 88.2% had dysesthetic macular skin lesions and 97.1% presented some peripheral nerve impairment, 71.9% with some degree of disability. All cases were multibacillary, confirming a late diagnosis. Anti-PGL-I results in the LG were higher than in the NLG ($p$<0.0001), while the RLEP-PCR was positive in 11.8% of the patients.

## Interpretation

Our findings within the penitentiary demonstrated a hidden prevalence of leprosy, although the individuals diagnosed were likely infected while living in their former communities and not as a result of exposure in the prison. The LSQ proved to be an important screening tool to help identify leprosy cases in prisons.

## Author summary

Leprosy looms still as a public health problem. Unfortunately, the drop in the number of leprosy cases in recent decades has been accompanied by general decline in the expertise of health care professionals to recognize signs and symptoms of leprosy, particularly at the early stages. The situation for individuals confined within a prison is likely worse because it would be rare for this population to receive a clinical exam for leprosy even in the case of obvious signs and symptoms. In this study, we used an active search strategy in the prison population providing them a Leprosy Suspicion Questionnaire (LSQ) with 14 simple questions about symptoms and signs related to leprosy. This questionnaire proved to be a simple, low-cost instrument for screening and identifying new leprosy cases among 1,400 prisoners. After responding to the questionnaire, they were evaluated clinically, and samples of blood and earlobe skin smears were collected for later laboratory anti-PGL-I IgM and RLEP-PCR assessment. Neurological symptoms ("anesthetized skin area" and "pain in nerves) were more commonly associated among the 34 new multibacillary cases diagnosed at penitentiary (rate 2.7%). The confinement time among leprosy patients (LG) was slightly longer than time of leprosy incubation indicating that the infection likely occurred before the imprisonment. Anti-PGL-I results in the LG were higher than in non-leprosy group and RLEP-PCR was positive in 11.8% of the patients, demonstrating that this disease is actually present in the confinement community. The application of the LSQ coupled with follow-up clinical exams and laboratory analysis are all actions with a high potential to reveal new leprosy cases in these confined communities.

## Introduction

Leprosy, caused by the obligate intracellular bacterium *Mycobacterium leprae* or *M. lepromatosis*, is a chronic contagious disease that affects peripheral nerves and skin [1,2]. Its transmission is thought to occur mainly through the upper airways of patients with a high bacillary load, properties that, depending on the specific bacillus-host interaction and the degree of endemicity in the environment [2]. *M. leprae* multiplies very slowly with a doubling time of around 13 days and for this reason the incubation period of the disease, on average, is 5 years [1].

Recent literature has shown hidden endemics in Brazil, even in areas that are supposed to be non-endemic areas [3,4]. This situation seems to be related to the low capacity of health professionals to carry out the diagnosis of leprosy, based essentially on the identification of clinical signs and symptoms [3,4]. For decades public policies for the diagnosis of leprosy have focused on the search for dermatological signs rather than neurological symptoms [3]. In addition, there have not been any studies on the potential of leprosy transmission in the Brazilian penitentiary population.

According the World Prison Brief it is estimated that more than 10.74 million people are held in penal institutions throughout the world, either as pre-trial detainees/remand prisoners or having been convicted and sentenced [5]. Brazil ranks 3[rd] among the countries with the largest prison population with 690,000 prisoners, behind the United States (2.1 million prisoners) and China (1.65 million) [5]. The overcrowding within prisons and their precariousness make the environment conducive to the spread of disease. Related to conditions of poor diet, lack of hygiene, physical inactivity, drug use, lack of preventative health screening, stress, among others, combine to make this population prone to illness [6]. The State of São Paulo has the largest prison population in Brazil, approximately 40% of the national total. In the literature there are only a few old leprosy epidemiological studies conducted of prison populations to determine if their risk is higher than the general population [7–9].

This work sought to evaluate the effectiveness of a new active search strategy for the diagnosis of leprosy based on both neurological symptoms and cutaneous signs, in addition to analyzing the clinical, immunoepidemiological aspects and follow-up of confined individuals in a Center of Penitentiary Progression (CPP).

## Methods

### Ethics statement

This study was approved by the Research Ethics Committee at the Clinics Hospital of Ribeirão Preto Medical School, University of São Paulo (protocol number 16620/2014 HCFMRP-USP, project MH Brazil). An informed written consent was obtained from every individual who agreed to participate in this study. All procedures involving human subjects comply with the ethical standards of the relevant national and institutional committees on human subjects' experimentation and with the Helsinki Declaration of 1975, as revised in 2008. Individuals who declined to participate in the study for any reasons were still evaluated clinically to provide them with the same level of care and treatment but their data were not used for analysis in this study.

### Type of study

This study was characterized as an observational cross-sectional comparative study.

### Characterization of place and action phases

The CPP of Jardinópolis is a prison unit for prisoners in the semi-open regime located in the interior of the State of São Paulo. CPP-Jardinópolis supposedly has a capacity of 1,080 prisoners, in a constructed area of 18,146 m$^2$, but it housed 1,864 individuals according to the census of September 27, 2016. In April 2016, the prison population was approximately 1,400 individuals.

The active leprosy search action in CPP was planned in phases from April to May 2016. The first was characterized by the distribution of a Leprosy Suspicion Questionnaire (LSQ) posing 14 simple questions about possible symptoms and signs related to leprosy (Fig 1) to the 1,400

| Leprosy Suspicion Questionnaire (LSQ) Center of National Reference in Sanitary Dermatology focusing on Leprosy Ribeirão Preto Clinical Hospital, Brazil Action: CPP-Jardinópolis | |
|---|---|
| **Name:** | |
| **Age:** | |
| **Check with "X" if there is change presence below** | |
| 1. Do you feel numbness in your hands and/or feet? | |
| 2. Tingling (pricking)? | |
| 3. Anesthetized areas in the skin? | |
| 4. Spots on the skin? | |
| 5. Stinging sensation? | |
| 6. Nodules on the skin? | |
| 7. Pain in the nerves? | |
| 8. Swelling of hands and feet? | |
| 9. Swelling of face? | |
| 10. Weakness in hands? | |
| 11. Hard to button shirt? Wear glasses? Write? Hold pans? | |
| 12. Weakness in feet? Difficulty wearing sandals, slippers? | |
| 13. Loss of eyelashes? | |
| 14. Loss of eyebrows? | |

**Fig 1. Leprosy Suspicion Questionnaire (LSQ).**

prisoners who were asked to answer within a week and deliver the completed survey to the nursing staff. A week later, individuals who elected to participate in the study received additional information about the project and filled out a signed consent form. An additional identification form was provided to collect demographic information about each subject. Then, subjects who voluntarily agreed to participate were referred for blood collection and to undergo a clinical dermatoneurological evaluation performed by 5 dermatologists, supported by nurses and biomedical doctors. Patients who received a leprosy diagnosis based on clinically well-established criteria as determined by at least two dermatologists were followed-up with a sample of a slit skin smear (SSS) collected for analysis of the presence of *M. leprae* DNA by standard PCR. Answers to LSQ were compiled and analyzed in Excel spreadsheets, as well as all clinical findings, including the anti-phenolic glycolipid I (PGL-I) IgM ELISA titer and PCR results.

## Diagnostic criteria for leprosy

The enrolled subjects underwent a standardized clinical dermatoneurological examination according to well-established World Health Organization guidelines. Leprosy diagnosis was made by the finding of at least one of the following signs/symptoms: a) definite loss of sensitivity and/or some dysautonomia in a hypochromic or reddish skin macule and/or b) a thickened or enlarged peripheral nerve with a respective loss of sensitivity and/or muscle weakness, and/or c) positive acid-fast bacilli detected in skin smears [10]. All leprosy diagnoses were certified by at least two experts. Considering that none of the classifications for leprosy include all of the clinical manifestations of leprosy, particularly those involving macular and pure neural forms, we classified the patients considering the guidelines adapted by Madrid (Congress of

Madrid 1953) [11] and the Indian Association of Leprology (IAL 1982) [12] classifications, as follows: indeterminate (I), polar tuberculoid (T), borderline (B), polar lepromatous (L) and pure neural leprosy (PNL); and broadly classified according to WHO operational criteria [PB (I and T) and MB (B and L)] [4]. Two groups were established as one consisting of individuals who were diagnosed with leprosy (leprosy group—LG) and the other consisting of other individuals (non-leprosy group—NLG).

## Assessment of anti-PGL-I titer by ELISA

An indirect ELISA was used to measure the anti-PGL-I IgM titer of all the serum samples tested at a 1:400 dilution using a protocol previously reported [13]. Briefly, ELISA microplate wells were coated overnight with synthetic PGL-I (12.5 ng/well ND-O-BSA) in 0.1M carbonate/bicarbonate pH 9.6 coating buffer (50 μl). After blocking (1% bovine serum albumin in phosphate buffered saline pH 7.2 with 0.05% Tween20, [1% BSA/PBS/T blocking solution]) for 2 hours, sera were diluted in this same blocking solution, and 75 μl add per well, incubated for 2 hours at room temperature. Then, the wells were washed six times with phosphate-buffered saline (PBS) with 0.05% Tween20 (PBS/T, wash buffer). Secondary peroxidase-conjugated anti-human IgM (1:20,000, Abcam, Cambridge, UK) was added for another 1h 30min incubation period. After this incubation, the wells were washed with PBS/T six times followed by the addition of 100 μl of substrate (3,3′,5,5′-tetramethylbenzidine; TMB). After 15 minutes at room temperature, 50 μl of stop solution ($H_2SO_4$, 1 M) was added.

Following, the optical density (O.D.) values were determined with an ELISA plate reader (Asys Expert Plus-Microplate Reader UK) at 450 nm. The cut-off for positivity was established at an O.D. of 0.295 based on the average plus three times the standard deviation of healthy subjects from a hyperendemic area as reported. The index of the sample was calculated by dividing their O.D. by 0.295, and indexes above 1.0 were considered positive.

## DNA extraction and RLEP amplification

Total DNA extraction of earlobe skin smear samples and at least one-elbow and/or lesion samples using the QIAamp DNA Mini Kit (Qiagen, Germantown, MD, Cat: 51306) was performed according to the manufacturers' protocol with minor modifications. Amplification of the *M. leprae* repetitive RLEP sequence (up to 37 copies are found within the genome) was achieved using GoTaq G2 Hot Start Taq Polymerase, Cat. M7401) according to a previously published protocol [14] using the primer pairs LP1 (5'-TGCATGTCATGGCCTTGAGG -3') and LP2 (5'-CACCGA TACCAGCGGCAGAA-3') described to amplify a 129-base pair fragment found in the genome.

## Statistical analysis

Statistical analysis to compare the differences between both groups were calculated using a non-parametric to the independent samples by the Wilcoxon-Mann-Whitney test using GraphPad Prism 5 software (GraphPad Software, San Diego, California, USA). The binomial test was used to calculate the difference between two percentages by Minitab software (Minitab LLC, State College, Pennsylvania, USA). The differences were considered statistically significant at conventional levels with $p < 0.05$.

## Results

### Analysis of leprosy suspicion questionnaire (LSQ) data

During the first week, the LSQ was distributed to all individuals in the prison population (n = 1,400 individuals). No specific questions mentioning the disease were given to the

**Table 1. Number of individuals ranked according to total signs and symptoms of leprosy marked on the LSQ in order of frequency (n = 187).**

| Q | Leprosy Suspicion Questionnaire (LSQ) | TOTAL (n) | % | NLG (n) | % | LG (n) | % |
|---|---|---|---|---|---|---|---|
| | Number of LSQ distributed | **1400**[*] | | - | - | - | - |
| | Number of LSQ returned | **896** | 71.7 | **865** | 96.5 | **31** | 3.5 |
| | Number of LSQ not-returned | 354 | 28.3 | **351** | 99.2 | **3** | 0.8 |
| | Number of individuals evaluated clinically | **1250** | 89.3 | **1216** | 97.3 | **34** | 2.7 |
| | Number of LSQ with some marking (LSQ+) | 187 | 20.9 | 169 | 90.4 | 18 | 9.6 |
| Q | **Symptoms and Signs (LSQ+)** | | | | | | |
| 1 | Do you feel numbness in your hands and/or feet? | 63 | 33.7 | 58 | 34.3 | 5 | 27.8 |
| 2 | Tingling (pricking)? | 77 | 41.2 | 71 | 42.0 | 6 | 33.3 |
| 3 | Anesthetized areas in the skin? | 30 | 16.0 | 22 | 13.0 | 8 | 44.4 |
| 4 | Spots on the skin? | 72 | 38.5 | 67 | 39.6 | 5 | 27.8 |
| 5 | Stinging sensation? | 43 | 23.0 | 37 | 21.9 | 6 | 33.3 |
| 6 | Nodules on the skin? | 41 | 21.9 | 38 | 22.5 | 3 | 16.7 |
| 7 | Pain in the nerves? | 60 | 32.1 | 52 | 30.8 | 8 | 44.4 |
| 8 | Swelling of hands and feet? | 16 | 8.6 | 13 | 7.7 | 3 | 16.7 |
| 9 | Swelling of face? | 10 | 5.3 | 9 | 5.3 | 1 | 5.6 |
| 10 | Weakness in hands? | 27 | 14.4 | 24 | 14.2 | 3 | 16.7 |
| 11 | Hard to button shirt? Wear glasses? Write? Hold pans? | 25 | 13.4 | 25 | 14.8 | 0 | 0.0 |
| 12 | Weakness in feet? Difficulty wearing sandals, slippers? | 9 | 4.8 | 9 | 5.3 | 0 | 0.0 |
| 13 | Loss of eyelashes? | 14 | 7.5 | 14 | 8.3 | 0 | 0.0 |
| 14 | Loss of eyebrows? | 13 | 7.0 | 13 | 7.7 | 0 | 0.0 |
| | Total number of answers | 500 | | 452 | | 48 | |
| | Mean answers/individual | 2.7 | | 2.9 | | 1.5 | |
| | Min | 1 | | 1 | | 1 | |
| | Max | 14 | | 14 | | 7 | |

Q: question number; n: number of checked questions; NLG: non-leprosy group; LG: leprosy group

[*] 150 prisoners were freed during the LSQ application week.

prisoners, and they were only requested to answer general questions about the signs and/or symptoms they reported feeling and to return the LSQs to the nursing staff. In total, 896 questionnaires were returned (64%), and 187 (20.9%) of these included being positive for one or more of the signs/symptoms as described in Table 1. Among the 187 respondents, each individual answered/filled 2.7 questions on average and their distribution by the number of marked answers in both groups is described in Table 1.

In total, 1,250 (89.3%) prisoners were evaluated in the morning, afternoon and also in the evening to clinically evaluate the individuals who work outside the unit. In the course of the action, 34 new cases were diagnosed, resulting in a new case detection rate of 2.7% within the CPP-Jardinópolis population.

Individuals were grouped in two groups for comparative analysis: those diagnosed with disease, Leprosy (LG, n = 34), and those with no clinical symptoms of leprosy, Non-Leprosy (NLG, n = 1216) groups.

Considering the strategy of applying the leprosy suspicion questionnaires, 18 patients were diagnosed among the 187 LSQ+ individuals, a new case detection rate (NCDR) of 9.6%, while 13 patients between 709 LSQ- (NCDR of 1.83%) and only 3 patients among 354 individuals who did not receive LSQ (NCDR of 0.85%). Thus, when analyzing the number of new cases found among the LSQ+ with the number of new cases obtained among the LSQ- and among

**Table 2. Demographic characterization of Leprosy and Non-Leprosy groups.**

| Groups | No Leprosy (n = 1216) | | Leprosy (n = 34) | | z | p |
|---|---|---|---|---|---|---|
| | n | % | N | % | | |
| Age (years) | | | | | | |
| Median | 30 | | 31.5 | | | 0.31 |
| Max | 76 | | 52 | | | |
| Min | 18 | | 19 | | | |
| Age range | n | % | N | % | z | p |
| < 20 | 44 | 3.6 | 2 | 5.9 | 0.69 | 0.49 |
| 20 \|--- 29 | 529 | 43.5 | 13 | 38.2 | 0.61 | 0.54 |
| 30 \|--- 39 | 455 | 37.4 | 13 | 38.2 | 0.10 | 0.92 |
| 40 \|--- 49 | 134 | 11.0 | 5 | 14.7 | 0.67 | 0.50 |
| ≥50 | 54 | 4.4 | 1 | 2.9 | 0.42 | 0.67 |
| State of birth | N | % | N | % | z | p |
| São Paulo | 1012 | 83.2 | 27 | 79.4 | 0.59 | 0.56 |
| Bahia | 31 | 2.5 | 2 | 5.9 | 1.2 | 0.23 |
| Alagoas | 4 | 0.3 | 2 | 5.9 | 4.62 | < 0.001 |
| Pernambuco | 20 | 1.6 | 1 | 2.9 | 0.58 | 0.56 |
| Minas Gerais | 47 | 3.9 | 1 | 2.9 | 0.28 | 0.78 |
| Paraná | 48 | 3.9 | 1 | 2.9 | 0.28 | 0.78 |
| Ceará | 10 | 0.8 | 0 | 0 | | |
| Piauí | 8 | 0.7 | 0 | 0 | | |
| Paraíba | 6 | 0.5 | 0 | 0 | | |
| Goiás | 5 | 0.4 | 0 | 0 | | |
| Maranhão | 4 | 0.3 | 0 | 0 | | |
| Mato Grosso | 4 | 0.4 | 0 | 0 | | |
| Rio Grande do Norte | 4 | 0.3 | 0 | 0 | | |
| Amazonas | 2 | 0.2 | 0 | 0 | | |
| Pará | 2 | 0.2 | 0 | 0 | | |
| Mato Grosso do Sul | 2 | 0.2 | 0 | 0 | | |
| Rio Grande do Sul | 2 | 0.2 | 0 | 0 | | |
| Sergipe | 1 | 0.1 | 0 | 0 | | |
| Rio de Janeiro | 1 | 0.1 | 0 | 0 | | |
| Distrito Federal | 1 | 0.1 | 0 | 0 | | |
| Acre | 1 | 0.1 | 0 | 0 | | |

those who did not receive LSQ, the chi-square statistical result was 40.3 (p<0.00001), with 6.4 relative risk and odds ratio 6.97.

The demographic characteristics of these two groups are described in Table 2. For the confinement time of individuals, LG presented a median of 39 months while the NLG was 36 months (not significant). The data about time in an open regime versus in a closed regime, including details on the number of prisoners per cell, are described in Table 3.

Concerning specific clinical aspects of the leprosy patients, 88.2% presented dysesthesia in macular lesional areas of the skin and 97.1% presented some impairment of peripheral nerves, so consequently 91.2% were classified as having the borderline leprosy (BL) form (Fig 2). Others clinical findings are described in Table 4. Electroneuromyography (ENMG) examination of the pure neural leprosy (PNL) cases presented an asymmetric and multifocal mononeuritis

**Table 3. Confinement time and number of prisoners per prison cell of LG and NLG.**

| | Total | NLG | | LG | | z | p |
|---|---|---|---|---|---|---|---|
| **Total confinement time (months)** | **n = 1,250** | **n = 1,216** | | **n = 34** | | | |
| Median | 36.0 | 36 | | 39 | | | |
| Max | 236 | 236 | | 204 | | | |
| Min | 0.1 | 0.1 | | 1 | | | |
| Confinement time in opened regime (months) | | | | | | | |
| In general jails | | | | | | | |
| Median | | 3 | | 1 | | | |
| Max | 43 | 43 | | 18 | | | |
| Min | 1 | 1 | | 1 | | | |
| (CPP-Jardinópolis) | | | | | | | |
| Median | 3 | 2 | | 1 | | | |
| Max | 43 | 43 | | 18 | | | |
| Min | 1 | 1 | | 1 | | | |
| Confinement time in closed regime (months) | | | | | | | |
| Median | | 20.0 | | 33.5 | | | |
| Max | 244 | 244 | | 192 | | | |
| Min | 0 | 0 | | 1 | | | |
| Distribution of confinement time (months) | n = 1,250 | n = 1,216 | % | n = 34 | % | z | p |
| <12 | 235 | 229 | 18.8 | 6 | 17.6 | 0.17 | 0.86 |
| 12 \|--- 24 | 227 | 223 | 18.3 | 4 | 11.8 | 0.98 | 0.33 |
| 24 \|--- 48 | 316 | 316 | 26.0 | 6 | 17.6 | 1.10 | 0.27 |
| 48 \|--- 96 | 369 | 358 | 29.4 | 11 | 32.3 | 0.37 | 0.72 |
| 96 \|--- 180 | 90 | 90 | 7.4 | 0 | | | |
| ≥180 | 11 | 11 | 0.9 | 0 | | | |
| No information | 2 | 2 | 0.2 | 0 | | | |
| Number of prisoners per prison cell | N | n | | n | | | |
| Median | 16 | 12 | | 16 | | | |
| Max | 200 | 168 | | 192 | | | |
| Min | 0 | 0 | | 2 | | | |

NLG: non-leprosy group; LG: leprosy group.

multiplex pattern. Among individuals from the NLG, 92% (920/1,000) had at least one BCG vaccination scar, while in the newly diagnosed case group it was 93.7% (30/32) as described in Table 4.

In the laboratory, 1,227 individuals from the CPP-Jardinópolis accepted to have their peripheral blood collected for serological analysis regarding the anti-PGL-I antibody. The 34 individuals diagnosed with leprosy also had samples of slit skin smear (earlobes, elbows and/or skin lesion) analyzed for the presence of *M. leprae* DNA using the molecular marker RLEP, registered at the Dermatology Laboratory of HCFMRP-USP.

Among the 1,250 individuals clinically evaluated, 23 refused blood collection for anti-PGL-I antibody. Considering all individuals tested for anti-PGL-I antibody (n = 1,227), 19.6% were positive in the NLG (234/1,193) with a median O.D. of 0.105, while 44.1% were positive in LG (15/34) with a median O.D. of 0.255 ($p<0.0001$). Also comparing the anti-PGL-I indices, 44.1% were positive in LG while 19.6% were positive in NLG (p = 0.005). Regarding the

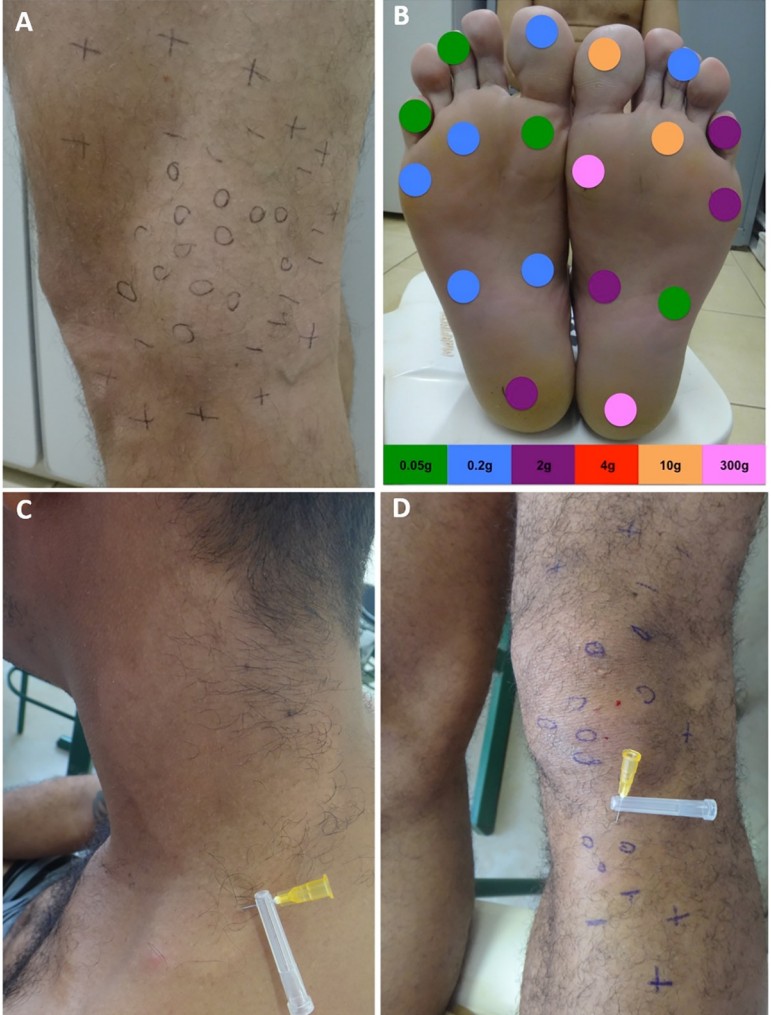

**Fig 2.** (a) Hypochromic, hypo-anesthetic macule on the left knee with loss of tactile sensitivity; (b) assessment of feet for touch sensitivity with the Semmes-Weinstein monofilaments test; (c) hypochromic and anesthetic to pain sensitivity, an anhidrotic macule on the neck; (d) multiple hypochromic and hypo-anesthetic to thermal, tactile and pain sensitivities, macule with localized irregular patches of circumscribed hair loss on the left lower limb. 0 (anesthetic point);—(hypoesthetic point); + (normoesthesic point).

positive anti-PGL-I results in LG, 13 (86.7%) were classified as borderline and 2 (13.3%) as pure neural leprosy, all with anti-PGL-I index higher than 1.5. The number of individuals with an anti-PGL-I index higher than or equal to 2.0 was higher in LG than in NLG (p<0.001). The anti-PGL-I values for the NLG and LG in O.D. are shown in Fig 3 and the distribution by indices for both groups is in Table 5.

A PCR assay was used to assess the presence of *M. leprae* DNA in SSS in all patients (four sites–two earlobes, elbow(s) and/or skin lesion), being positive in only 4/34 (11.8%) patients, among which 3 of these also had high titers of anti-PGL-I antibody with indexes >1.5.

## Follow-up of patients

All patients were followed up by a dermatologist and a leprosy physician from May 2016 to July 2017 during the clinical and therapeutic follow-up for 12 months, with reported clinical manifestations described in Table 6. In relation to the decrease in the number of patients being

**Table 4. Clinical characterization of CPP-Jardinópolis patients regarding the percentage of positivity to the clinical criteria used for the diagnosis of leprosy (n = 34) and data about BCG scar in Non-leprosy group distributed according the anti-PGL-I results.**

| Clinical criteria | Yes (n = 34) | % | anti-PGL-I + (n = 15) | % | anti-PGL-I— (n = 19) | % | z | p |
|---|---|---|---|---|---|---|---|---|
| Dysesthesia hypochromic macular skin lesions | 30 | 88.2 | 13 | 86.7 | 17 | 89.5 | 0.25 | 0.80 |
| Tactile sensitivity | 5 | 14.7 | 2 | 13.3 | 3 | 15.8 | 0.20 | 0.84 |
| Tactile + pain sensitivities | 6 | 17.6 | 5 | 33.3 | 1 | 5.3 | 2.13 | 0.03 |
| Tactile + thermal sensitivities | 2 | 5.9 | 2 | 13.3 | 0 | 0 | | |
| Thermal + tactile + pain sensitivities | 17 | 50.0 | 5 | 33.3 | 12 | 63.2 | 1.73 | 0.08 |
| Dysesthetic areas (without cutaneous signs) | 11 | 32.4 | 2 | 13.3 | 9 | 47.4 | 2.11 | 0.03 |
| Tactile sensitivity | 1 | 2.9 | 0 | 0 | 1 | 5.3 | | |
| Tactile + pain sensitivities | 1 | 2.9 | 0 | 0 | 1 | 5.3 | | |
| Thermal + tactile + pain sensitivities | 10 | 29.4 | 2 | 13.3 | 8 | 42.1 | 1.83 | 0.07 |
| Localized irregular patches of circumscribed hair loss | 25 | 73.5 | 12 | 80.0 | 13 | 68.4 | 0.76 | 0.45 |
| Altered nerves on palpation (enlargement and/or pain and/or electric shock-like pain) | 33 | 97.1 | 15 | 100 | 18 | 94.7 | 0.90 | 0.37 |
| Endogenous histamine test | | | | | | | | |
| Incomplete | 17 | 50 | 8 | 53.1 | 9 | 47.4 | 0.35 | 0.73 |
| Not performed | 17 | 50 | 8 | 53.1 | 9 | 47.4 | 0.35 | 0.73 |
| Sweat test (alizarin test) | | | | | | | | |
| Hypohidrosis and/or anhidrosis in islets | 4 | 11.8 | 1 | 6.67 | 3 | 15.8 | 0.82 | 0.41 |
| Leprosy classification | | | | | | | | |
| Borderline | 31 | 91.2 | 15 | 100 | 16 | 84.2 | 1.61 | 0.11 |
| PNL | 3 | 8.8 | 1 | 6.7 | 2 | 10.5 | 0.39 | 0.70 |
| WHO operational criteria | | | | | | | | |
| Multibacillary | 34 | 100 | 15 | 100 | 19 | 100 | 0.0 | 1.0 |
| WHO impairment grading | | | | | | | | |
| Grade 0 | 9 | 28.1 | 4 | 26.7 | 5 | 26.3 | 0.01 | 0.99 |
| Grade 1 | 20 | 62.5 | 9 | 60.0 | 11 | 57.9 | 0.12 | 0.90 |
| Grade 2 | 3 | 0.9 | 1 | 6.7 | 2 | 10.5 | 0.39 | 0.70 |
| Not evaluated | 2 | - | 1 | - | 1 | - | | |
| BCG scar | | | | | | | | |
| 0 | 2 | 11.8 | 1 | 6.7 | 1 | 5.3 | 0.17 | 0,87 |
| 1 | 29 | 85.3 | 13 | 86.7 | 16 | 78.9 | 0.59 | 0.56 |
| ≥2 | 1 | 2.9 | 0 | 0 | 1 | 3.1 | | |
| Not evaluated | 2 | - | 1 | - | 1 | - | | |
| BCG scar—NLG | | | | | | | | |
| 0 | 80 | | 20 | 10.3 | 60 | 7.5 | 1.77 | 0.08 |
| 1 | 899 | | 171 | 87.7 | 728 | 90.4 | 2.38 | 0.02 |
| ≥2 | 21 | | 4 | 2.0 | 17 | 2.1 | 0.16 | 0.87 |
| Not evaluated | 216 | - | 51 | - | 165 | - | | |

PNL: pure neural leprosy; WHO: World Health Organization.

followed up during their treatment period, this was due to the penitentiary regime as being characterized as semi-open, in which most individuals are at the end of their sentence with their discharge from prison during the follow-up. For these discharged individuals, medical reference documents were made available to the receiving units in their municipalities according to the address provided, in addition to immediate communication being available with the

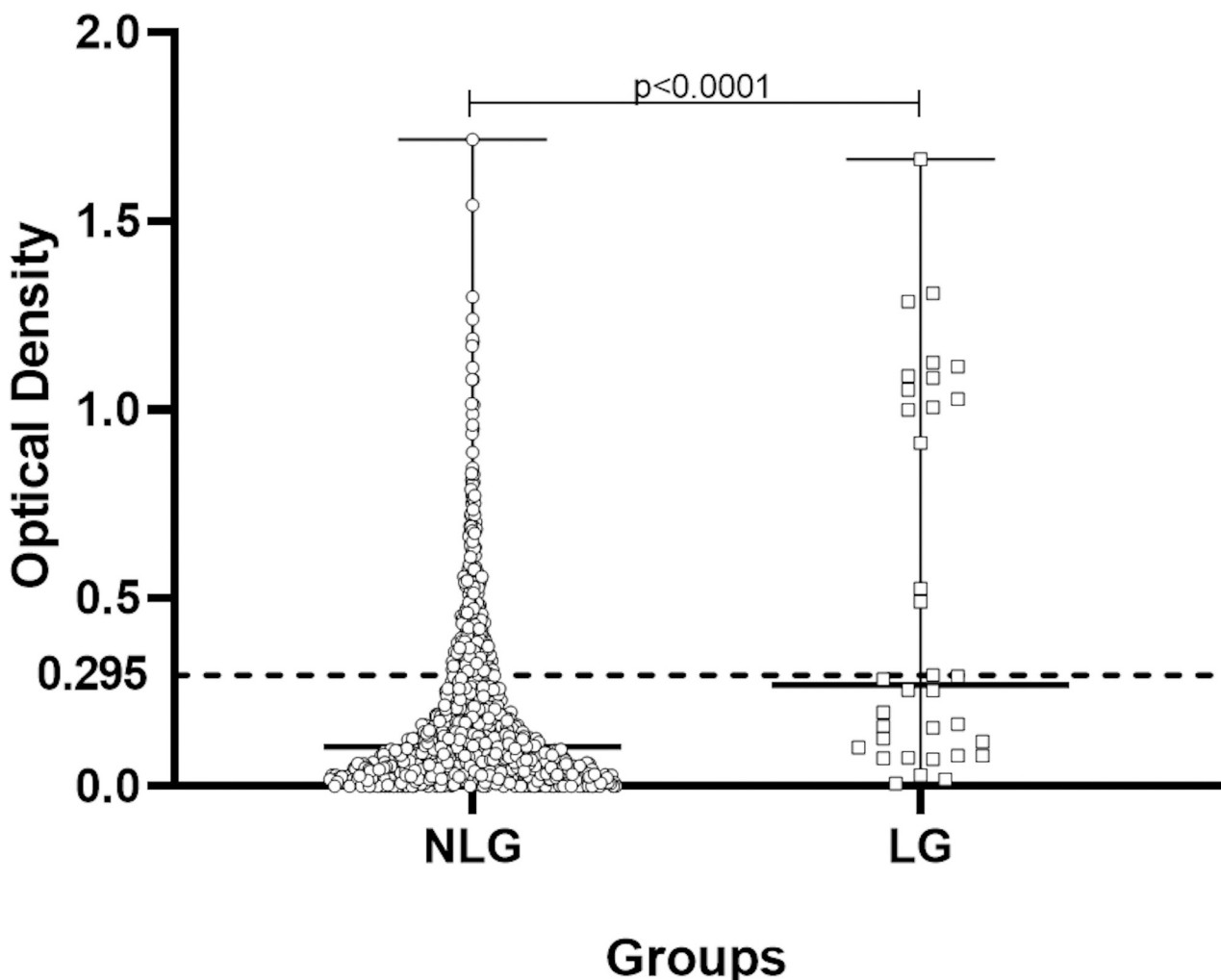

**Fig 3. Analysis by optical density (O.D.) of anti-PGL-I ELISA titer of individuals from Non-Leprosy Group and patients from Leprosy Group.**

state's epidemiological surveillance network. In addition, it should be noted that two of patients escaped during a CPP-Jardinópolis rebellion that occurred in the fifth supervised dose.

The beginning of the third and of the seventh months of treatment, a greater frequency of neurological and dermatological symptoms and signs evolved, ranging from improvement, conditions slightly worsening and/or additional changes as described in Table 6.

**Table 5. Results of anti-PGL-I antibody measurements (anti-PGL-I index; cut off 0.295) among CPP-Jardinópolis tested prisoners (n = 1,227).**

|  | Total |  | Non-Leprosy Group |  | Leprosy Group |  | z | p |
|---|---|---|---|---|---|---|---|---|
|  | n = 1,227 | % | n = 1,193 | % | n = 34 | % |  |  |
| anti-PGL-I < 1 (negative) | 978 | 79.7 | 959 | 80.4 | 19 | 55.9 | 3.50 | 0.005 |
| anti-PGL-I ≥ 1 (positive) | 249 | 20.3 | 234 | 19.6 | 15 | 44.1 | 3.50 | 0.005 |
| 1.0 |--- 1.5 | 127 | 10.4 | 127 | 10.6 | - | - | - | - |
| 1.5 |--- 2.0 | 56 | 4.6 | 53 | 4.4 | 3 | 8.8 | 1.21 | 0.23 |
| ≥ 2.0 | 66 | 5.4 | 54 | 4.5 | 12 | 35.3 | 7.84 | < 0.001 |

**Table 6. Clinical data and follow up in the leprosy patients treated.**

| Supervised monthly MDT-dose | 1st | 2nd | 3rd | 4th | 5th | 6th | 7th | 8th | 9th | 10th | 11th | 12th |
|---|---|---|---|---|---|---|---|---|---|---|---|---|
| Number of followed-up patients | 29 | 25 | 21 | 13 | 12 | 11 | 9 | 6 | 4 | 2 | 2 | 2 |
| Signs & Symptoms | | | | | | | | | | | | |
| Improvement of neurological symptoms (cramps, numbness and/or tingling) | | P14 | P26 P13 | | P28 P25 | | P06 P20 | | | | P25 | |
| Improvement of neural pain on palpation | | P14 | P13 | P28 | | | | | | | | |
| Improvement of esthesiometry | | | P26 | | P25 | | P18 | | | | | |
| Improvement of WHO impairment grading | | | | | P25 | | P06 P20 | | | | | |
| Appearing of ichthyosis in islets in the lower limbs | | | P26 P13 | | | | P06 P26 P20 | P32 | P32 | P25 | | |
| Improvement of skin sensitivity | | | P27 | | | | | | | | | |
| Conjunctival eye irritability/dryness | | | P32 | | | | P06 | | P32 | | | |
| Skin lesions became more evident | | | P09 | | | | | | | | | |
| Skin lesions became less evident | | | P31 | P31 | | | P11 | | | | | |
| Appearing of numbness and cramps | | | P27 | | | | | | | | | |
| Improvement of cutaneous reflex by endogenous histamine test | | | P31 | | | | | | | | | |
| Neuritis | | | | | | | P11 | | | | | |
| Pain on neural palpation | | | | | | | | P18 | | | | |
| Total patients with manifest symptoms/signs | 0 | 1 | 6 | 2 | 2 | 0 | 5 | 3 | 1 | 1 | 1 | |

P: patient.

## Discussion

The CPP of Jardinópolis, linked to the São Paulo state Penitentiary Administration Secretariat is a semi-open prison model that is quite interesting for the study of diseases related to confinement. The prisoners in this facility were mostly completing the final period of their sentence, most of whom had already served a period of time in other facilities. The prisoners that were transferred to this facility were mainly those with records of good behavior and low risk, and were provided better opportunities for medical care not only related to leprosy but also to other morbidities.

According to the Brazil Penitentiary Department, in relation to the total sentence time of individuals in the prison population, 14.2% had sentences between 2 and 4 years, 32.9% of prisoners have sentences between 4 and 8 years, and 24.9% between 8 and 15 years [15]. In line with the national data on length of sentence for the prison population, our sample showed a higher number of individuals serving a prison time of 4 to 8 years (30%). However, the second largest group had a prison time between 2 and 4 years (25%), that is, less time than the second group described for national detentions.

Infectious diseases in the incarcerated population can become a major health problem within prisons because of overcrowding conditions and other factors related to poor nutrition and stress and they are an important target for prevention. There are few studies in the literature on leprosy within prisons [7–9], unlike tuberculosis [6,16], the latter which has clearly established data on its risks in this population. Therefore, much is unknown and can only be speculated about the risks of leprosy transmission among confined individuals. This paper adds to the scarce literature on the incidence of leprosy in incarcerated individuals.

Considering that the incubation period of the leprosy is thought to be between 3 to 5 years, and with the known medians of confinement time of each group, 36 months in NLG and 39

months in LG, even with the high number of prisoners per cell during their past history of confinement, around 16, we can infer that leprosy transmission did not occur within the prison unit but instead probably occurred in their respective communities of origin, that is, prior to imprisonment. If this is true, it is interesting to note that 79% of the patients are from cities in the state of São Paulo, a warning sign because it is officially a state where leprosy has been in control since 2006, with a prevalence under 1 case per 10,000 inhabitants [4,17]. The most prevalent age group inside the prison coincides with the age range that is most affected by leprosy, that being between 20 and 40 years of age, corroborating that found in leprosy literature in the general population, which highlights the importance of evaluating this type of population in confinement.

There was an excellent rate of completion of the questionnaire by the inmates of this facility (89.3%), with an average number of checked responses of 2.7 questions per participant. Interestingly, among the 31 new cases of leprosy who responded, only 1.5 responses were found on average, which was slightly lower than in NLG. It is worth mentioning that the symptoms and signs described in the questionnaire are not specific because our goal was to screen the population about the risk of leprosy, but we also found other diseases or conditions, such as diabetes, HIV, hepatitis infection, carpal tunnel syndrome and others. Although none of the participants were informed that the nature of the survey was about leprosy, the specific questions asked were those about signs and symptoms routinely found in leprosy patients, such as the neurologic symptoms related to the presence of tingling, numbness in the hands and/or feet, pain in the nerves, sensation of pricks and needles, and hypopigmented or red and scaly spots on the skin, responses that were found in more than 30% of questionnaires in addition to the main known symptom, "anesthetized area in the skin".

In assessing the signs and symptoms in patients, 30/34 (88.2%) had dysesthetic skin lesions, highlighting the loss of three sensations (thermal/tactile/pain) in 52%, while 58% presented dysautonomia due to localized irregular patches of circumscribed hair loss. The diagnosis was also evaluated by the use of the endogenous histamine test with incomplete reaction in 17/34 (50%) and by visualizing the loss of the ability of areas of skin to sweat using the alizarin test [18] in 4/34 (11.8%) of the patients. ENMG defined the pattern of the peripheral neural impairment for three of the PNL cases. Although ENMG is more sensitive than the clinical examination for the detection of nerve impairment [19,20], we highlight that these patients had some physical disability at their diagnosis, turning out to be a late diagnosis. Intradermal vaccination with Bacillus Calmette-Guérin (BCG) is compulsory for all infants in Brazil and since the 1960s studies have shown that evidence of a BCG scar is associated with leprosy protection, with boosters given to immediate contacts of new cases of leprosy [21,22]. According to Brazilian guidelines for leprosy surveillance, there is no reference to prisoners in the definition of contacts for leprosy.

All of the leprosy cases were categorized as multibacillary, with 33/34 (91.2%) of the patients having multiple peripheral nerves with enlargement, pain and/or electric shock-like pain on palpation and 23/32 (71.9%) with some degree of disability in the diagnosis, confirming the late diagnosis of these patients. In addition, reliance solely on cursory examination of skin lesions reveals its limitations since the importance of including a thorough peripheral neurologic examination performed by palpation of nerves, a simplified neurological evaluation and esthesiometry using Semmes-Weinstein monofilaments to assess loss of sensation often reveals early signs of nerve damage, that is, loss of sensation in areas without obvious skin lesions and inflammation characteristic of leprosy.

Anti-PGL-I serology was positive in 15 patients (44.1%), with 12 (35.3%) of these having anti-PGL-I indices ≥2.0, a higher percentage than in NLG with 54 individuals (4.5%). Additionally, another concern is that 234/1,193 (19.6%) of individuals in the NLG have a positive

anti-PGL-I titer, which is very high for a state and region with an officially controlled endemic disease. Brasil *et al* [23] demonstrated that a positive anti-PGL-I titer is a biomarker for *M. leprae* infection and carries around an 8-fold higher risk of disease progression and our data showed by the significant percentage of anti-PGL-I indices higher than or equal to 2 among patients (LG) compared to that found in the NLG, that this assay presented a significant potential for screening in the prison community and probably can be useful in the community in general. Clearly, the high rate of anti-PGL-I among prisoners signals the need for periodic reassessment and new screening (possibly examining earlobe *M. leprae* colonization by PCR) with clinical re-evaluation and follow-up aimed at real control of the disease within CPP-Jardinópolis. Unfortunately, due to the unit's prison progression characteristics, we were unable to perform follow-up on these NLG individuals with high anti-PGL-I titers.

Health systems in prisons are often not amenable to managing diseases or other health conditions effectively, especially those that require chronic and regular assessments, as with leprosy. Regular follow-up consultations with physicians require security and logistic concerns in and outside of the unit, and it may raise the morbidity impact on health [24]. Throughout the one-year treatment period, two dermatologists and leprologists of the research team assessed leprosy patients monthly. During the follow-up, it should be noted that improvements in clinical manifestations in response to multidrug therapy (MDT) began to appear in the 2nd month and more significantly in the 3rd month in around 30% of patients, increasing to around 56% improvement by the 7th month. The signs and symptoms most frequently showing improvement include: appearing of ichthyosis in islets (32%), improvement of neurological symptoms such as cramps, numbness and tingling (28%), decrease of loss of sensation as measured by esthesiometry (12%), improvement of the WHO impairment disability grade (12%) and improvement of neural pain on palpation (12%). Only one patient had a reactional episode. These neurological manifestations presented in the evolution follow the well-documented infectious *M. leprae* etiology of neuropathy in response to antibacterial treatment.

Regarding the clinical and therapeutic follow-up of patients, a significant benefit in providing clinical referrals with photographs of the lesions and the patient's clinical history to the CPP health network, which worked well as a very useful and highly appreciated instrument for patient follow-up, avoiding issues of rejecting patient diagnoses that we have encountered elsewhere [25], which thankfully did not occur with CPP-Jardinópolis patients under our medical supervision at the prison.

The limitations of our study were the routine lack of patients due to limited freedom offered by the CPP regime and, consequently, it was not possible to revisit individuals with high anti-PGL-I indexes and a complete one-year clinical follow-up of released patients and evaluation of their household contacts was also not possible due the long distance to their respective hometowns.

## Conclusions

The active search for new leprosy cases at the penal institution of CPP-Jardinópolis demonstrates a hidden prevalence and that the disease may be an unrecognized problem among individuals in confinement, although data show that the acquisition of the disease likely occurred in their respective communities of origin and not within this penitentiary.

The LSQ proved to be an important screening tool for new leprosy cases within CPP-Jardinópolis and its general detection rate (89.3% of the total prison population were evaluated) was 2.7%, while among the LSQ+ the NCDR was 9.6%, meaning a relative risk of 6.4. In addition, it was also an instrument of health education, reviving the signs and symptoms of leprosy in the collective consciousness in the population and health professionals within this facility.

High anti-PGL-I indices ($\geq 2$) seem to demonstrate the potential to also potentially be an additional tool for leprosy screening in and probably out of the prison population.

Finally, in view of the number of cases found in CPP-Jardinópolis, the success of the effectiveness of the LSQ, a simple, low-cost instrument for screening and possibly identifying new leprosy cases, the importance of the clinical and therapeutic follow-up of patients, this study seeks to alert the Penitentiary Administration Departments and the Health Secretariats regarding leprosy surveillance.

## Author Contributions

**Conceptualization:** Fred Bernardes Filho, Marco Andrey Cipriani Frade.

**Data curation:** Fred Bernardes Filho, Natália Aparecida de Paula, Marco Andrey Cipriani Frade.

**Formal analysis:** Fred Bernardes Filho, Jaci Maria Santana, Regina Coeli Palma de Almeida, Glauber Voltan, Natália Aparecida de Paula, Marcel Nani Leite, Claudia Maria Lincoln Silva, Camila Tormena, Lean Basoli, Joelma Menezes, Wilson Marques, Jr, Marco Andrey Cipriani Frade.

**Funding acquisition:** John Stewart Spencer, Marco Andrey Cipriani Frade.

**Investigation:** Fred Bernardes Filho, Jaci Maria Santana, Regina Coeli Palma de Almeida, Glauber Voltan, Natália Aparecida de Paula, Marcel Nani Leite, Claudia Maria Lincoln Silva, Camila Tormena, Lean Basoli, Joelma Menezes, Moises Batista da Silva, John Stewart Spencer, Wilson Marques, Jr, Norma Tiraboschi Foss, Marco Andrey Cipriani Frade.

**Methodology:** Fred Bernardes Filho, Marco Andrey Cipriani Frade.

**Project administration:** Fred Bernardes Filho, Marco Andrey Cipriani Frade.

**Resources:** Fred Bernardes Filho, Natália Aparecida de Paula, Marcel Nani Leite, Claudia Maria Lincoln Silva, Camila Tormena, Lean Basoli, Joelma Menezes, Moises Batista da Silva, John Stewart Spencer, Wilson Marques, Jr, Marco Andrey Cipriani Frade.

**Supervision:** Marco Andrey Cipriani Frade.

**Visualization:** Fred Bernardes Filho, Marco Andrey Cipriani Frade.

**Writing – original draft:** Fred Bernardes Filho, Marco Andrey Cipriani Frade.

**Writing – review & editing:** Fred Bernardes Filho, Natália Aparecida de Paula, Moises Batista da Silva, John Stewart Spencer, Wilson Marques, Jr, Norma Tiraboschi Foss, Marco Andrey Cipriani Frade.

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
