## [Decision Letter · Decision Letter 0]

20 Jul 2020

Dear Dr Frade,

Thank you very much for submitting your manuscript "Leprosy in the prison population: an active search strategy, epidemiological and a prospective clinical analysis" for consideration at PLOS Neglected Tropical Diseases. As with all papers reviewed by the journal, your manuscript was reviewed by members of the editorial board and by several independent reviewers. In light of the reviews (below this email), we would like to invite the resubmission of a significantly-revised version that takes into account the reviewers' comments. 

We cannot make any decision about publication until we have seen the revised manuscript and your response to the reviewers' comments. Your revised manuscript is also likely to be sent to reviewers for further evaluation.

Sincerely,

Carlos Franco-Paredes

Associate Editor

Christine Petersen

Deputy Editor

Reviewer's Responses to Questions

**Key Review Criteria Required for Acceptance?**

**Methods**

-Are the objectives of the study clearly articulated with a clear testable hypothesis stated?

-Is the study design appropriate to address the stated objectives?

-Is the population clearly described and appropriate for the hypothesis being tested?

-Is the sample size sufficient to ensure adequate power to address the hypothesis being tested?

-Were correct statistical analysis used to support conclusions?

-Are there concerns about ethical or regulatory requirements being met?

Reviewer #1: There was a good objective and it gave a lot of intersting and locally needed side information. Population was well desrcibed could even be less. 

I have no problems with ethical requirements. But in Manilla( World leprosy congress some had)

Reviewer #2: mentioned need to be elaborated fully.

Reviewer #3: -Are the objectives of the study clearly articulated with a clear testable hypothesis stated? Yes. However, the authors should be more clear with the writing of the objectives. 

-Is the study design appropriate to address the stated objectives? Yes.

-Is the population clearly described and appropriate for the hypothesis being tested? No.

-Is the sample size sufficient to ensure adequate power to address the hypothesis being tested? Yes. However the authors did not show the sample size calculi.

-Were correct statistical analysis used to support conclusions? No, it was insuficient.

-Are there concerns about ethical or regulatory requirements being met? Yes.

**Results**

-Does the analysis presented match the analysis plan?

-Are the results clearly and completely presented?

-Are the figures (Tables, Images) of sufficient quality for clarity?

Reviewer #1: As mentioned one table particularly need adaptation (table 6)

Reviewer #2: Tables need further clarity

Reviewer #3: -Does the analysis presented match the analysis plan? Yes.

-Are the results clearly and completely presented? No. More associations may be explored.

-Are the figures (Tables, Images) of sufficient quality for clarity? No. Data and statistical test should be inserted.

**Conclusions**

-Are the conclusions supported by the data presented?

-Are the limitations of analysis clearly described?

-Do the authors discuss how these data can be helpful to advance our understanding of the topic under study?

-Is public health relevance addressed?

Reviewer #1: Conclusions are discussed and relevant

Reviewer #2: Conclusion that LSQ questionnaire was useful was not substantiated.

Reviewer #3: -Are the conclusions supported by the data presented? Yes, however, the conclusion is too long.

-Are the limitations of analysis clearly described? No.

-Do the authors discuss how these data can be helpful to advance our understanding of the topic under study?

-Is public health relevance addressed?Yes.

**Editorial and Data Presentation Modifications?**

Reviewer #1: I think it can be more to the point and more concise

Reviewer #2: (No Response)

Reviewer #3: No comments.

**Summary and General Comments**

Reviewer #1: Dear Authors,

I was very interested to read your paper since I had heard in Manilla about it. but I had difficulties reading it. Being not an English speaker I felt some of the sentences have to much Portuguese influence, making it not very clear what was actually meant. One of the authors is a native English speaker. It would be good when he has a second look at the English and for the main author I like it more concise and much less wordy. It should be shortened. 

I was impressed by the sensitivity of the questionnaire and learned from the results of the questions.. 

Interesting was the observation that the leprosy was most likely not acquired in the prison despite the living conditions mentioned. The anti PGL-1 was high in some of the non-patients. Could it be that that there is continues contact with this antigen either in prison or since they are also allowed to go out to their home environment. Then it would be worthwhile to look at their contacts at home. 

I agree that it is frightening that there are so many not diagnosed leprosy patients from Sao Paulo state. 

Some remarks:

1 Title: in a prison Check the second half: epidemiological and a prospective clinical analyses (it is to much)

6 would say in a prison population

8 A questionnaire 

39 is imprisonment not better then reclusion? 

46 it is bon ton to include M. lepromatosis . For me it is not necessary but some like it.

47 infectious and contagious is a repetition just another word

56 there have been studies but may be not published . In other countries there are quite a number ao. In the Indian leprosy journal . Look in Pubmed.

66 see above, it is in the Indian literature

110 just a remark: without smears or equivalent you may underdiagnose leprosy. You have mostly prevented this by PCR just mention this.

117 low resistant T by some are classified by BT and could be in BP.

143 was it one or both earlobes? Doing only the earlobes you may miss positivity. 

144 just a question the 3 patients who did not participate were they different from the other patients.

Table 1 do you have an explanation why the NLG have more positive complaint answers?

Table 2 just to remark you have included here also the patients who did not want to participate. 

Table 3 is this table needed? There is no significant difference between the groups can that not be just mentioned.? 

Table 4 My complements for the clinical testing.

220 Now I noticed SSS were from 4 sites. Were they mixed together before testing? 

Table 6 In the second line I see No of patients with sign and symptoms. Are this the number of patients seen? Than you followed only 2 for a year. The last line with manifest symptoms the first column comes to 0. When I later see improvement in some patients . I think the table should be more clear. In some patients I see deterioration but later no improvement afterwards. Not seen the patient back? Does this table contribute? 

The “Ichthyosis” is due to clofazimine? Or due to decreased inflammation?

243 Why is a semi open prison to study diseases related to confinement better then a closed one? 

259 I see now that you mentioned some of the Indian papers.

271 which doe actually highlight what. You just corroborate. 

284 Dysautonomies : other word? I understand what you meant to say.

286 Alizarin test needs explanations or reference. To date it is mostly used in Brazil in other programs it is not used anymore while it is a sensitive test.

286 With shock on palpation you mean Tinel sign?

311 It was also not possible to test their contacts at home? 

318-326 This needs a clear understandable table 6

Please make this interesting article more to the point.

What is the difference between these authors contribute equally to this work and

 & the ones contributed also equally?

Reviewer #2: An interesting paper on active leprosy case search in prison population, Providing good data with the help of questionnaire and clinical examination supported by lab tests.The fact that out about 1250 prisoners, 34 MB leprosy patients were newly diagnosed based on clinical / neurological examination with lab support is an important finding in this closed community. The paper needs to highlight and discuss this aspect more fully.

Reviewer #3: The authors investigated, by means of Leprosy suspicion questionaire (LSQ), a hidden prevelence of leprosy in a prison population, important findings, since that, this individuals are not assessed for no one infectious disease as a routine.

Bellow are my comments by subsection:

Abstract

In line 5, the word “investigation” instead “search” may be better for this contexto of diagnosis.

In the section Methods of the abstract, authors should state the type of study. It seems to be a cross-sectional for leprosy diagnosis by means of Leprosy suspicion questionaire (LSQ). It could be important define the range of data collection even if the authors have stated they performed a –follow-up time. The follow-up time make part of cross-sectional for investigation of cases during the declared time.

On the other hand, if authors think they carried out a prospective cohort, what I can not believe, the time zero and time-to-evento/outcome (signs and symptoms) must be declared. The proof that this study is a cross-sectional, refers to the fact the authors used the prevalence term.

Introduction

From line 68 to 69 authors quoted they “evaluate the effectiveness of a new active search strategy...”. I disagree with the authors, because they evaluate a prison individuals, using a Leprosy Suspicion Questionnaire (LSQ), which help them to diagnosis and characterizing clinically and epidemiologically this sample, besides of serological and PCR investigations. Effectiveness refers to application of clinical trials or mathematical models to prove better performance using numerical data.

Subjects and methods

From line 74 to 82, the section “Ethics, consent and permissions” generally is placed in the end of the Methodology. Although this section is considered important, readers would rather start the reading with the section called type of study or sample.

I suggest to authors start the section "subjects and Methods" with a subsection “ Type of study and sample” stating if cross-sectional, retrospective cohort, prospective or another type of study. Moreover, authors should state who is the sample, that is, who is the LG and NLG. The total number of patients per group may be declared if authors agree with this suggestion.

The authors should write a new subsection intitled “Phases of data collection” placing the text from line 89 to 101.

The subsection “Diagnostic criteria for leprosy” must be maintained, except to the text from line 117 to 119 which is more interesting if inserted in the new subsection "Type of study and sample or similar".

A subsection, “Statistical Analysis” should be created on this section “Subjects and Methods”.

Authors should drive attention in the statistics text placing better structure to express what tests were applyed in the manuscript. 

For instance, The Mann-Whitney U test is used to comparing differences between two independent groups regardind to sum of ranks. New statistical tests should be explored as suggested in this revision, for example, Binomial test. 

What was the statistical software used for this analysis?

Results

In the table 2, authors should calculate a p-value for each line in relation to the variable “Age range” with the goal to confirming that there are no diferences (or there are differences) among each age group. I recommend the use of Binomial test to give evidence that there is/or there is no difference between two percentages. The same statistical analysis should be applied to the state of birth, since that if there is difference between two percentages for each state, it will indicate a possible association encompassing NLG or LG with state of birth. In the lines where the percentage is equal zero there are no p-values. The statistical test should be pointed out in the section "statistical analysis" together Wilcoxon-Mann-Whitney test.

In the table 3, if the data are non-parametric, there is no reason to show mean, but only median and maximum and minimum values. Why do the authors did not compare the groups medians related to the times exposed in table 3? It is useful indicating the statistical differences between the two groups.

In table 4, It could be more intersting to making a table keeping the line variables associated with clinical criteria, however, inserting 2 columns dividing the LG into seropositives and seronegatives PGL-1 variable. This suggestion may help you find out new findings and confirm potential associations. A recommended test may be Binomial or Chi-square test. The first one is more intersting due to compare two percentages per line.

Figure 3 – Authors should provide a self-explanatory legend figure and indicating the name of statistical test - Mann-Whitney test -, besides the "U",given by the test, to becaming clear for readers how big is the difference between the groups.

In table 5, I suggest to the authors provide a statistical test to verify if there was difference between NLG versus LG in relation to PGL1 percentuge results. 44,12% of positivity in LG is different statistically from 19,6% from NLG ?

The binomial test may be useful.

The p-value and the Z value provided by the Binomial test should be shown at the two last columns of the table 5.

Another option would be the authors disscuss indicators as sensibility, specificity, false positive/false negative rates.

 From line 224 to 233, this information could be placed in the section “Subjects and Methods”.

The table 6 showed a follow up time in reference to clinical data of treated patients. However, this data could be intersting whether authors had performed a follow up associated with an outcome and provided us a survival analysis. On the other hand, these informations is not important in this context.

Conclusions

The conclusions are too long. Authors must sumarize the findings.

PLOS authors have the option to publish the peer review history of their article (what does this mean?). If published, this will include your full peer review and any attached files.

Reviewer #1: No

Reviewer #2: No

Reviewer #3: No
---

## [Decision Letter · Decision Letter 1]

13 Oct 2020

Dear Dr Frade,

Thank you very much for submitting your manuscript "Leprosy a the prison population: an active search strategy and a prospective clinical analysis" for consideration at PLOS Neglected Tropical Diseases. As with all papers reviewed by the journal, your manuscript was reviewed by members of the editorial board and by several independent reviewers. The reviewers appreciated the attention to an important topic. Based on the reviews, we are likely to accept this manuscript for publication, providing that you modify the manuscript according to the review recommendations. 

Sincerely,

Carlos Franco-Paredes

Associate Editor

Christine Petersen

Deputy Editor

Reviewer's Responses to Questions

**Key Review Criteria Required for Acceptance?**

**Methods**

-Are the objectives of the study clearly articulated with a clear testable hypothesis stated?

-Is the study design appropriate to address the stated objectives?

-Is the population clearly described and appropriate for the hypothesis being tested?

-Is the sample size sufficient to ensure adequate power to address the hypothesis being tested?

-Were correct statistical analysis used to support conclusions?

-Are there concerns about ethical or regulatory requirements being met?

Reviewer #1: I am of the opinion that the methods are good

Reviewer #3: I am satisfied with the changes inserted into the text by authors in this section.

**Results**

-Does the analysis presented match the analysis plan?

-Are the results clearly and completely presented?

-Are the figures (Tables, Images) of sufficient quality for clarity?

Reviewer #1: The analyses matches the majority of the plan. 

Results are mostly clear

Reviewer #3: After authors agreed with recommendations about to insert statistical analysis in tables 2, 3,4,5, the information have became rich for readers.

**Conclusions**

-Are the conclusions supported by the data presented?

-Are the limitations of analysis clearly described?

-Do the authors discuss how these data can be helpful to advance our understanding of the topic under study?

-Is public health relevance addressed?

Reviewer #1: The conclusions are supported by the data. Limitations are well described . They discuss their data well but could consider other possibilities as well. Public Health is adressed.

Reviewer #3: No comments.

**Editorial and Data Presentation Modifications?**

Reviewer #1: I think it should be more clear not only how many where indicated but also how many were missed by the questionaire

Reviewer #3: (No Response)

**Summary and General Comments**

Reviewer #1: Dear Authors,

In my opinion the paper has improved. The English has improved but could be more concise,. But hat is my opinion not a native English speaker. 

I however still have some comments and questions. 

One is you had 896 LSQ’s returned 1250 you evaluated. How many of the 31 (34) patients diagnosed were not in the picture after analysing the data of the questionnaire. This will show how good the questionnaire is in indicating leprosy. How many were suspect? This is important for the conclusion that the LSQ is an important screening tool.(not only it found 2.7%.

69: Look at this sentence that the Anti-PGl1 was higher in The LG demonstrates that this disease is one of many. I think that does it not indicate.

77;Transmission etc check that sentence, it talks abut the exit of bacilli from patients, but the entré can be direct via respiratory track or indirect via the environment to the respiratory track and also through the skin.

86; I would leave “In addition” out. It introduces something completely different.

98: the work sought to evaluate a new search strategy. I thought that is the Questionnaire. But were is it in the results? As diagnostic test/ You look at the different symptoms which is very interesting. But what as test generally? (see 189 -204) 

In table 3 it is not clear to me how many per cell. Later in the text you mention it somewhere.(302: 16)

Table 4: I translate the findings to Ridley Jopling quite a number are BT explaining the low number of PGL positive among your MB’s . Also the low number of PCR positive ones. 

I understood that the median time they were in this prison was about 3 years. So most will be infected before. But most were in other prisons already many years, thus they may be infected in these prisons and not at home. Thus I doubt 303-304. Just consider it again. 

348: positive AntiPGL1 is according to me not a biomarker for infection. It only shows contact with the antigen. The majority of the bacilli after contact are dead and 80% will never develop leprosy but may be anti- PGL1 positive .( Naafs Indian J Med Res 147, January 2018, pp 1-3 and Info Hansen 2020 https://en.infohansen.org/blog/morbus-hansen)

But the test is good for epidemiological purposes, thus I agree 352.

I agree fully with 371-375. 

In 385 you say it clearly the acquisition was not in this penitentiary. But it may be in another prison which was or was not in their home communities.

386 in conclusions you make clear the diagnostic rate was 2.7% in the questionnaire screened population. But I would like to see this also in the results, but I may have missed it. How many did it miss? 

Further I want to complement you with all the hard work you have done.

Reviewer #3: (No Response)

PLOS authors have the option to publish the peer review history of their article (what does this mean?). If published, this will include your full peer review and any attached files.

Reviewer #1: No

Reviewer #3: No
---

## [Editor Report · Decision Letter 2]

23 Oct 2020

Dear Dr Frade,

We are pleased to inform you that your manuscript 'Leprosy a the prison population: an active search strategy and a prospective clinical analysis' has been provisionally accepted for publication in PLOS Neglected Tropical Diseases.

Best regards,

Carlos Franco-Paredes

Associate Editor

Christine Petersen

Deputy Editor

---

## [Editor Report · Acceptance letter]

24 Nov 2020

Dear Dr Frade,

We are delighted to inform you that your manuscript, "Leprosy a the prison population: an active search strategy and a prospective clinical analysis," has been formally accepted for publication in PLOS Neglected Tropical Diseases.

Best regards,

Shaden Kamhawi

co-Editor-in-Chief

Paul Brindley

co-Editor-in-Chief
